# Circadian Rhythm Perturbation Aggravates Gut Microbiota Dysbiosis in Dextran Sulfate Sodium-Induced Colitis in Mice

**DOI:** 10.3390/nu16020247

**Published:** 2024-01-12

**Authors:** Joseph Amara, Tarek Itani, Joelle Hajal, Jules-Joel Bakhos, Youakim Saliba, Saied A. Aboushanab, Elena G. Kovaleva, Nassim Fares, Alicia C. Mondragon, Jose Manuel Miranda

**Affiliations:** 1Laboratoire de Recherche en Physiologie et Physiopathologie, Pôle Technologie Santé, Faculté de Médecine, Université Saint Joseph, Beirut 1104 2020, Lebanon; joseph.amara@laumcsjh.com (J.A.); joelle.hajal2@usj.edu.lb (J.H.); youakim.saliba@usj.edu.lb (Y.S.); 2Laboratoire de Microbiologie, Faculté de Pharmacie, Université Saint Joseph, Beirut 1104 2020, Lebanon; tarek.itani@usj.edu.lb; 3Institute of Chemical Engineering, Ural Federal University Named after the First President of Russia B. N. Yeltsin, Mira 19, Yekaterinburg 620002, Russia; sabushanab@urfu.ru (S.A.A.); e.g.kovaleva@urfu.ru (E.G.K.); 4Laboratorio de Higiene, Inspección y Control de Alimentos, Departamento de Química Analítica, Nutrición y Bromatología, Campus Terra, Universidade da Santiago de Compostela, 27002 Lugo, Spain; alicia.mondragon@usc.es

**Keywords:** DSS-induced colitis, dysbiosis, circadian rhythm, bacterial diversity, microbiota, gut

## Abstract

Circadian rhythm disruption is increasingly considered an environmental risk factor for the development and exacerbation of inflammatory bowel disease. We have reported in a previous study that nychthemeral dysregulation is associated with an increase in intestinal barrier permeability and inflammation in mice with dextran sulfate sodium (DSS)-induced colitis. To investigate the effect of circadian rhythm disruption on the composition and diversity of the gut microbiota (GM), sixty male C57BL/6J mice were initially divided to two groups, with the shifted group (*n* = 30) exposed to circadian shifts for three months and the non-shifted group (*n* = 30) kept under a normal light–dark cycle. The mice of the shifted group were cyclically housed for five days under the normal 12:12 h light–dark cycle, followed by another five days under a reversed light–dark cycle. At the end of the three months, a colitis was induced by 2% DSS given in the drinking water of 30 mice. Animals were then divided into four groups (*n* = 15 per group): sham group non-shifted (Sham-NS), sham group shifted (Sham-S), DSS non-shifted (DSS-NS) and DSS shifted (DSS-S). Fecal samples were collected from rectal content to investigate changes in GM composition via DNA extraction, followed by high-throughput sequencing of the bacterial 16S rRNA gene. The mouse GM was dominated by three phyla: Firmicutes, Bacteroidetes and Actinobacteria. The Firmicutes/Bacteroidetes ratio decreased in mice with induced colitis. The richness and diversity of the GM were reduced in the colitis group, especially in the group with inverted circadian rhythm. Moreover, the GM composition was modified in the inverted circadian rhythm group, with an increase in *Alloprevotella*, *Turicibacter*, *Bacteroides* and *Streptococcus* genera. Circadian rhythm inversion exacerbates GM dysbiosis to a less rich and diversified extent in a DSS-induced colitis model. These findings show possible interplay between circadian rhythm disruption, GM dynamics and colitis pathogenesis.

## 1. Introduction

Inflammatory bowel diseases (IBD), including ulcerative colitis (UC) and Crohn’s disease (CD), are chronic and relapsing diseases arising from an interaction between genetic and environmental factors [1,2,3]. The identification of over two hundred IBD-associated susceptibility genes underscores the complexity of these diseases, with some genes implicated in mediating host responses to the gut microbiota (GM) [4].

The human GM is a vast ecosystem comprising more than one hundred trillion different microbial organisms, including bacteria, viruses, fungi and protozoa [5]. Within this diverse microbial community, more than one thousand bacterial species inhabit the human GM, and they mostly belong to four phyla—Firmicutes, Bacteroidetes, Proteobacteria and Actinobacteria. Bacteroidetes and Firmicutes dominate the GM in healthy adults, maintaining a symbiotic relationship with the human host [6]. This symbiosis is profitable to the host in several ways, including through protection against pathogens, immuno-regulation, nutrition and metabolism [7]. Intestinal dysbiosis, a disruption of the normal intestinal flora, is increasingly associated with several pathologies, including IBD [8,9], obesity [10,11], diabetes [12], autoimmune disease and allergies [13,14].

Despite the important heterogeneity of IBD patients, the type of disease, the inflammation location (ileum, colon, rectum) and the disease’s activity, several prominent and common changes were observed in their GM. Past authors found, in both CD and UC patients, a decrease in bacterial diversity associated with Firmicutes depletion and increases in Proteobacteria [15] and, more specifically, in *Enterobacteriaceae* [16,17,18]. Moreover, proinflammatory bacteria, especially mucosa-associated bacteria (*Escherichia coli*) and mucolytic bacteria (*Ruminococcus gnavus* and *Ruminococcus torques*), were increased in the gut of IBD patients [19,20]. This increase was shown to affect intestinal permeability, disturb gut biodiversity and enhance immunological and inflammatory responses, exacerbating intestinal inflammation in IBD patients [21]. Additionally, Gαi2 subunit-deficient mice, which develop ulcerative colitis-like disease and are more susceptible to IBD, had decreased relative abundance levels of *Lactobacillus* and *Bacteroides* with a high degree of colitis [22].

Drug-induced colitis in murine models has become the model of choice to study, in a controlled experimental setup, the interactions between GM and the host. The dextran sulfate sodium (DSS) model has been extensively used to induce colitis in mice [23]. Under these conditions, the mouse GM showed a decrease in gut microbiome diversity associated with epithelial permeability increase, systemic and intestinal inflammation exacerbation and immune dysregulation, major axes implicated in the pathogenesis of IBD [24].

Shift working with circadian rhythm disruption is considered an environmental risk factor for the development of IBD in genetically predisposed patients [2,3]. Intriguingly, sleep disturbance was shown to be associated with immune deregulation, inflammation [25] and an increase in intestinal permeability [26]. These findings were supported by observational human and experimental animal studies [27,28].

The current study builds upon our prior work, demonstrating that chronic circadian rhythm disruption aggravated DSS-induced colitis in mice. This aggravation was correlated with an increase in intestinal barrier permeability with fecal and intestinal calprotectin elevation [29]. Our new objective is to gain a deeper understanding of the mechanism through which circadian rhythm disruption aggravates pre-existing colitis by studying the interaction between the GM and circadian variations.

Despite the known environmental and genetic factors influencing IBD, the combined influence of DSS and circadian rhythm perturbation on the composition of the GM remains inadequately explored. A wide heterogenicity and even some recent contradictions were found in their conclusions [30,31]. Therefore, this study aims to assess the effect of nychthemeral disruption on the diversity and composition of the GM in DSS-induced colitis in mice with inverted circadian rhythms.

## 2. Material and Methods

### 2.1. Animals

This study was conducted on 8-to-10-week-old male C57BL/6J mice (*n* = 60). Animals were housed at a stable temperature (25 °C) and humidity (50 ± 5%) and exposed to a 12:12 h light–dark cycle. They were fed ordinary rodent chow and had free access to tap water. An acclimation period of at least one week under these conditions preceded the start of this study.

### 2.2. Experimental Protocol

The present study was approved by the Ethical Committee of the Saint Joseph University of Beirut (Protocol code FPH86, approved on the 12 December 2018). Following an acclimation period of one week, the mice were divided into two groups: shifted and non-shifted. The non-shifted group (*n* = 30) adhered to a consistent light–dark cycle, while the shifted group (*n* = 30) was exposed to a cyclic reversal of the circadian rhythm every 5 days for 3 months. The circadian shift was accomplished by alternating between a 12:12 h light–dark cycle for five days (lighted from 7 P.M. till 7 A.M.) and its reversed counterpart (lighted from 7 P.M. till 7 A.M.) [29,32]. Subsequently, the mice were divided into four groups (*n* = 15 per group): Sham-NS (sham non-shifted), Sham-S (sham shifted), DSS-NS (DSS non-shifted) and DSS-S (DSS shifted). Colitis was induced by adding 2% DSS into the drinking water for 7 consecutive days, followed by a 3-day recovery period before sacrifice [33,34,35] (Figure 1).

### 2.3. Extraction and Purification of Total DNA from Feces

For sacrifice, animals were anesthetized with ketamine (75 mg kg^−1^; Interchemie, Waalre, Holland) and xylazine (10 mg kg^−1^; RotexMedica, Trittau, Germany). The pedal withdrawal reflex was tested to ensure an adequate depth of anesthesia. Fecal samples were collected (100 mg to 150 mg) from rectal content to investigate changes in GM composition at the species level. It was notable that all mice, irrespective of the study group, were sacrificed at the same time of day. DNA extraction was performed using the Zymo Quick-DNA™ Fecal/Soil Microbe Miniprep (Zymo Research, Irvine, CA, USA) according to the manufacturer’s instructions. The bead-beating step was achieved using a homogenizer (Biospec, Bartlesville, OK, USA) twice for 2 min. The concentration and quality of DNA were measured using a Nanodrop D100 Spectrophotometer (Nanodrop Technology, Wilmington, DE, USA).

### 2.4. High-Throughput Sequencing of Bacterial 16S rRNA Gene

The microbial community was assessed via the high-throughput sequencing of the bacterial 16S rRNA gene through the GeT-PlaGe platform in INRAE Toulouse (France) using Illumina MiSeq technology. The V3V4 region was amplified from purified DNA with the primers F343 (CTTTCCCTACACGACGCTCTTCCGATCTTACGGRAGGCAGCAG) and R784 (GGAGTTCAGACGTGTGCTCTTCCGATCTTACCAGGGTATCTAATCCT) using 30 amplification cycles with an annealing temperature of 65 degrees (an amplicon of 510 bp). Single multiplexing was performed using a homemade 6 bp index, which was added to R784 during a second PCR with 12 cycles using a forward primer (AATGATACGGCGACCACCGAGATCTACACTCTTTCCCTACACGAC) and reverse primer (CAAGCAGAAGACGGCATACGAGAT-index-GTGACTGGAGTTCAGACGTGT). The resulting PCR products were purified and loaded onto Illumina MiSeq cartridges (San Diego, CA, USA) according to the manufacturer’s instructions. The quality of the run was checked internally using PhiX, and then each pair-end sequence was assigned to its samples with the help of the previously integrated index. Each pair-end sequence was assembled using Flash software v1.2.6 (Magoc 2011) using an at least 10 bp overlap between the forward and reverse sequences, allowing 10% mismatch. The lack of contamination was checked with a negative control during the PCR, using water as a template. The quality of the stitching procedure was controlled using 4 bacterial samples that were run routinely in the sequencing facility in parallel to the current samples.

### 2.5. Bioinformatics Analysis

Sequences were analyzed and normalized with the pipeline FROGS (Find Rapidly Operational Taxonomic Units (OTUs) with Galaxy Solution) [36]. PCR primers were removed, and sequences with sequencing errors in the primers were excluded. Reads were clustered into OTUs using the Swarm clustering method. Chimeras were removed, and 273 OTUs were assigned at different taxonomic levels (from phylum to species) using the RDP classifier and NCBI Blast+ on the Silva_123_16S database.

A total of 12,289 reads were randomly selected for each sample to normalize the data. The sequences were aligned using Clustal Omega 1.1.0 via the profile alignment option in Sea View 4.5 [37]. Neighbor joining trees, as well as maximum-likelihood trees using PhyML 3.1, were built to assess identifications [38].

### 2.6. Statistical Analysis

The GM of the mice in all groups was analyzed using high-throughput sequencing (average number of reads ± SEM = 13,735 ± 2775). Microbial diversity analyses were performed by clustering sequence tags into groups of defined sequence variation. *α*-Diversity measurements (observed OTUs, Chao 1, Shannon diversity index or SDI and inverted Simpson index) and *β*-diversity measurements (Jaccard, Bray–Curtis, UniFrac and weighted UniFrac) were analyzed using a blocked analysis of variance. The relative abundance of bacteria was compared with a MULTINOVA using the Jaccard and unweighted UniFrac similarity measures to construct distance metrics using QIIME 2 (Quantitative Insights Into Microbial Ecology 2) v 2019.7. All analyses were conducted using the R programming language in FROGS.

## 3. Results

The mouse GM was dominated by three phyla: Firmicutes, Bacteroidetes and Actinobacteria (Figure 2). The Firmicutes/Bacteroidetes ratio was 1.78, 1.68, 1.36 and 1.36 for the Sham-NS, Sham-S, DSS-NS and DSS-S groups, respectively. The Firmicutes/Bacteroidetes ratio decreased in tandem with induced colitis.

At the family level, the GM was mainly colonized by Lactobacillaceae, Lachnospiraceae, Muribaculaceae, Ruminococcaceae, Erysipelotrichaceae, Rikenellaceae and Prevotellaceae (Figure 3). In addition, the obtained results showed that Lactobacillaceae increased in mice with a reversed circadian cycle (sham-NS: 28% of sequences vs. sham-S: 47.5%; DSS-NS: 10.2% vs. DSS-S: 20.4%), Lachnospiraceae decreased in mice with a reversed circadian cycle (sham-NS: 27.2% vs. sham-S: 6.8%), Muribaculaceae colonized all groups at similar levels, and Ruminococcaceae increased in mice with induced colitis and a normal circadian cycle (DSS-NS: 16.3%). Finally, Erysipelotrichaceae mainly colonized mice with induced colitis (DSS-NS: 4.3% and DSS-S: 5.96%). All groups were poorly colonized by Proteobacteria and not colonized by Akkermansiaceae.

At the genus level, we found more *Lactobacillus* when the circadian rhythm was reversed (Sham NS 28% vs. Sham S 47.5% and DSS-NS 10.2% vs. DSS-S 20.5%). Furthermore, *Lachnospiraceae* NK4A136 was decreased in mice with a normal circadian rhythm (Sham NS 8.6% vs. Sham S 2.7%). On the other hand, mice with induced colitis were colonized more by *Ruminococcaceae UCG-14* and *Alloprevotella* than mice without colitis. In addition, *Turicibacter* was mainly detected in mice with induced colitis, regardless of circadian rhythm (DSS-NS 4.3% and DSS-S 5.9%). *Helicobacter* was found in low percentages in the Sham-S (0.6%) and DSS-NS (0.8%) groups but was almost undetectable in mice with reversed circadian rhythm (Figure 4).

### 3.1. Effects of Circadian Rhythm and Induced Colitis on Microbial Diversity

Samples collected from the four groups were analyzed and compared in terms of *α*-diversity metrics (observed OTU richness, Chao-1, Shannon diversity index and inverted Simpson). All the metrics (except the inverted Simpson) differed significantly between mice with normal and reversed circadian rhythm, with or without induced colitis (Figure 5).

OTU richness was significantly higher (*p* = 0.0005) in the Sham NS group. The *α*-diversity metrics SDI and Chao-1 decreased significantly (*p* = 0.00003 and *p* = 0.045, respectively) in mice with reversed circadian rhythm and with or without colitis compared to Sham NS mice. These results show that the number of species increased significantly in mice with normal circadian rhythm and without colitis, while reverting circadian rhythm and inducing colitis decreased bacterial richness and diversity. The influence of these factors (circadian rhythm and colitis) reduced bacterial richness and diversity.

### 3.2. Effects of Circadian Rhythm and Induced Colitis on Microbial Communities

Following the binning of the sequences into operational taxonomic units (OTUs) based on a 97% sequence identity, comparisons were made using principal coordinate analysis (PCoA) based on Jaccard and UniFrac distances (*β*-diversity). Each sample corresponding to microbial communities from mice with normal circadian rhythm and reversed circadian rhythm clustered tightly and separated on the second principal axis (P2). Samples from mice with no colitis or induced colitis clustered tightly and separated on the first principal axis (P1). A developmental shift in the mouse microbial community with an inverse circadian rhythm was detected on the second principal component axis (Figure 6).

When samples obtained from mice with induced colitis were compared with the Sham NS group, microbial communities from mice that subsequently developed colitis (in blue and violet, Figure 6) were distinct from those of mice without colitis (in red and green, Figure 6). The continued analysis of the microbial communities from these mice reinforced the observed divergence of the colitis group when compared to the controls, particularly on the first principal axis (P1). Unweighted UniFrac analysis and Jaccard analysis revealed that the difference in *β*-diversity was explained by reverting the circadian rhythm and/or inducing colitis.

## 4. Discussion

The obtained results show, for the first time, that the inversion of the circadian rhythm and the presence of DSS-induced colitis were associated with intestinal dysbiosis. Mice with isolated disrupted circadian rhythm have a different GM composition and a significant reduction in diversity compared to sham mice. In addition, mice with both circadian rhythm inversion and DSS-induced colitis have a different composition pattern, with a significant reduction in the diversity of the bacterial gut community (Figure 6).

Gut bacterial diversity was reduced in the stool and intestinal tissues of patients with CD and UC and in several DSS animal model studies [15,30,39,40,41]. Here, the comparison of GM diversity between disrupted circadian rhythm groups and normal circadian rhythm groups revealed interesting modifications. The difference in *α*- and *β*-diversities shows that circadian disruption is associated with a lower variety of GM. This disparity underlines the role of the circadian rhythm in the composition of the microbiome, and, moreover, it may reflect the additional influence of the nychthemeral changes, especially in DSS and DSS inverted circadian rhythm, compared to the group with colitis and normal exposure to light. Our findings endorse other those of studies concerning the role of normal light cycles in the composition and maintenance of the GM [28]. These diversity changes may have important functional relevance. Gut dysbiosis was found in DSS-induced colitis with or without light disturbances and was associated with the alteration of epithelial junctional protein expression, thus perturbing gut permeability, a major component of gut defense [29].

Circadian disruption alone did not induce significant changes in the physiology or GM composition. Clinical and inflammatory manifestations were significantly present only in the subgroup of mice suffering from DSS-induced colitis, particularly in the DSS-S group [29]. These observations are consistent with the “two hit” hypothesis, where nychthemeral perturbations, even if they modify the GM richness and composition, are not sufficient alone to promote pathology. A second metabolic or chemical hit is needed to reveal the negative additional action of circadian perturbation [42,43].

With respect to the GM and circadian rhythm, there is a bidirectional relationship. Thus, the circadian clock influences the composition of the GM; likewise, the GM regulates the circadian rhythm. Studies are currently focusing on the disruption of this bidirectional axis [12]. Indeed, the human GM has been shown to play a key role in the development of brain function and, consequently, in stress-related diseases and neurodevelopmental disorders [12].

Accordingly, with previous observations, this work shows that the mouse microbiome was dominated by three major phyla: Firmicutes, Bacteroidetes and Actinobacteria. The stool collected from mice with an inverted circadian rhythm and DSS-induced colitis revealed a lower abundance of Firmicutes and a higher abundance of Bacteroidetes and Actinobacteria populations. Notably, the Firmicutes/Bacteroidetes ratio was decreased in the DSS group compared to the control group. Gut mucosal inflammation was associated with a decreased Firmicutes/Bacteroidetes ratio and an increase in Proteobacteria and Actinobacteria [44]. Members of the Bacteroidetes phylum have already been considered markers of disease onset [45] or inducers of colitis in mouse models [46]. Bacterial expression is also influenced by light exposure: Bacteroidetes peak several hours after the beginning of the dark phase, and Firmicutes peak around the beginning of the light phase [47]. The Firmicutes/Bacteroidetes ratio has been extensively studied in human and mouse GM. After seven years of age, the ratio between Bacteroidetes and Firmicutes is relatively stable, while a disturbance might contribute to metabolic syndrome, such as obesity and type-2 diabetes. In obesity model studies, this ratio decreased in diabetic mice [48] and increased in obese mice [49]. Furthermore, a significant decrease was also found in some patients with UC or CD compared to controls and considered to be a dysbiosis indicator in IBD [50,51]. Interestingly, our results revealed that a reduction in the Firmicutes/Bacteroidetes ratio was also present in the group of mice with circadian rhythm perturbation, even without colitis. This finding supports a previous observation in ClockΔ19 mice fed alcohol with modification of the Firmicutes/Bacteroidetes ratio [43]. In addition, it was demonstrated that the probiotic *Saccharomyces boulardii* reduced the Firmicutes/Bacteroidetes ratio in DSS-treated mice, simultaneously reducing gut dysbiosis [30]. However, the administration of the probiotic Lactobacillus plantarum in DSS-treated mice increased the Firmicutes/Bacteroidetes ratio [31]. These differences in the Firmicutes/Bacteroidetes ratio could be related to the age of the subjects [52] and their diet and body mass index [53]. Despite these divergences, the modification of the Firmicutes/Bacteroidetes ratio in mice exposed to circadian rhythm indicates the role of the molecular clock in determining the GM composition.

At the family level, we noticed a significant increase in *Ruminococcaceae*, *Erysipelotrichiaceae*, *Streptococcaceae, Staphylococcaceae* and *Rikenellaceae* in the stool of mice with colitis, with a particular increase in *Bacteroides*, *Streptococcus*, *Turicibacter* and *Alloprevotella* at the genus level. *Lactobacillus* and butyrate-producing *Lachnospiracea* species decreased with the induction of colitis. Two genera of the Bacteroidetes phylum were significantly more highly expressed in the colitis group: *Bacteroides* and *Alloprevotella*. An additive enrichment was also noted with the inversion of the circadian rhythm. Our findings concurred with the results of two recent studies showing increased *Alloprevotella* concentrations in mice with DSS [54] and obese mice [55], with a decrease in these concentrations in the groups treated with melatonin, a regulator of the circadian rhythm, heightening the role of the circadian clock in the composition of the GM. Our findings also concurred with the observations of Bishehsari et al. [56], who noticed an increase in *Bacteroides* populations while studying the carcinogenic effect of circadian disruption. The concentration of the genus *Desulfovibrio*, a sulfate-reducing Proteobacteria frequently found in the stools of UC patients and associated with a reduction in mucosal thickness [24], was significantly higher in the group of mice with colitis and inverted circadian rhythm—another argument for the additive effect of circadian inversion on gut defense mechanisms.

The composition of the Firmicutes phylum was also modified in mice with colitis and with circadian rhythm modifications: *Turicibacter* concentrations were increased in the colitis group. Its role in IBD physiopathology remains unclear. Some studies described a decreased concentration of *Turicibacter* in colitis [57], while others described an increased concentration in murine DSS-induced colitis [58]. We noticed a supplementary increase in *Turicibacter* populations in the DSS groups and circadian shifts. Its presence may be related to increased TNFα expression [59] and CD8 T-cell depletion, which are two major actors in the inflammatory response [60]. Their expression is also increased in Bmal-1KO mice, a major gene regulator in the molecular circadian clock [47].

*Lactobacillus* concentrations significantly decreased in the colitis group, but the circadian rhythm inversion surprisingly induced an increase in their concentrations. *Streptcoccus* concentrations were elevated in the DSS group and had an inverted circadian rhythm. *Streptococcus* populations also increased in inflammation [61] and decreased in UC patients with normalized sleep patterns [62]. Our findings are coherent with these previous observations.

Some controversy was found in our findings concerning *Lactobacillaceae*, *Ruminococcacea* and *Lachnospiracea*. *Lactobacillus* concentration increased in the stools of mice with inverted circadian rhythm, with or without induced DSS colitis. There are still various reports about changes in the concentrations of *Bifidobacterium* and *Lactobacillus* in IBD patients. Their use as probiotics in colitis mouse models has confirmed their experimental benefit in treating drug-induced colitis. The predominant observations are that their concentrations are decreased in IBD patients and they have anti-inflammatory effects that prevent intestinal inflammation by increasing the expression of tight junction proteins and mucin secretion and producing AMP, which combat pathogenic invasion [63]. However, few studies performed on CD patients, especially in active disease, have shown increased concentrations of *Bifidobacterium* and *Lactobacillus* [64,65], or even *Faecalibacterium prausnitzii*, in fecal samples [66,67,68]. Some strains of the *Lactobacillaceae* family, *L. rhamnosus* GG, *L. rhamnosus* KLSD, *L. helveticus* IMAU70129 and *L. casei* IMAU60214, may exacerbate DSS-induced colitis [69] and are involved in macrophage activation and the early initiation of inflammation in humans via mechanisms that implicate the synthesis of proinflammatory mediators, such as cytokines and reactive oxygen species, and participation in signaling cascades, such as the NF-κB and TLR2 pathways [70]. These divergences in the literature are probably due to the phylogenetic diversity of the *Lactobacillus* genus, which includes more than 100 species [71]. The microbial diversity analysis shows that reverting circadian rhythm and inducing colitis decreased bacterial richness and diversity. These results are in accordance with a Chinese study by Hu et al., where the authors found decreased microbial species richness and diversity and a shift in microbial community with a decreased proportion of Firmicutes [72].

## 5. Limitations

Our study offers valuable insights, but certain limitations should be considered. Age-atching randomization was not performed among the different mouse groups, despite our knowledge of the influence of age on the GM composition [52,53]. The rhythmicity of the different bacterial expressions related to eating or light exposure was also not taken into consideration. A further elaborative study of this rhythmicity may elucidate some of the controversies found in our study concerning *Lactobacillaceae*, *Lachnospiraceae* and *Ruminococcaceae* expression.

While we observed dysbiosis associated with circadian rhythm disruption and induced colitis, our study lacked functional data and mechanistic insights. The inclusion of shotgun functional data in future research would enhance our understanding of the direct implications of observed microbial changes without diminishing the significance of our dysbiosis observations. Despite this limitation, our study contributes to the broader understanding of circadian rhythm perturbation and gut dysbiosis, offering key observations that merit further investigation. Future research incorporating age-matched randomization, addressing bacterial rhythmicity and integrating functional data will provide a more comprehensive understanding of the intricate interplay between circadian rhythm, GM and colitis.

Caution is warranted when extrapolating our findings to human subjects, and the translational relevance to clinical scenarios requires further investigation. Nonetheless, the robustness of our observed dysbiosis in the murine model lays the groundwork for future studies that explore the applicability of these insights in human contexts. This work guides future directions for research and clinical applications, emphasizing the possible need for probiotic interventions in circadian-disrupted individuals suffering from IBD [28].

## 6. Conclusions and Clinical Implications

Our study elucidates the intricate relationship between circadian rhythm disruption, induced colitis and GM composition. Notably, we observed significant alterations at the phylum, family and genus levels, shedding light on the nuanced interplay between these factors. The Firmicutes/Bacteroidetes ratio decreased with induced colitis, reflecting changes in microbial populations. Furthermore, our findings highlighted changes at the family and genus levels with considerable fluctuations. Microbial diversity analysis revealed significant differences between mice with normal and reversed circadian rhythms, with or without induced colitis. Our study underscored a reduction in bacterial richness and diversity associated with circadian disruption and colitis. *β*-Diversity analysis demonstrated distinct microbial communities in mice with normal circadian rhythm, reversed circadian rhythm, induced colitis and induced colitis with reversed circadian rhythm. Notably, the “two-hit” hypothesis was supported, indicating that circadian disturbances alone were insufficient to induce significant changes, with clinical manifestations more pronounced in the colitis subgroup. Despite these insights, our study acknowledges limitations, such as age disparities, unexplored bacterial rhythmicity and the absence of functional data. Future research incorporating age-matched randomization, addressing rhythmicity and integrating functional data is crucial for a comprehensive understanding. While caution is needed for extrapolating findings to humans, our study provides a foundation for exploring probiotic interventions in circadian-disrupted individuals. In essence, our research contributes valuable knowledge to the evolving fields of circadian biology and gut health, paving the way for further investigations into clinical applications and preventive strategies.

## Figures and Tables

**Figure 1 nutrients-16-00247-f001:**
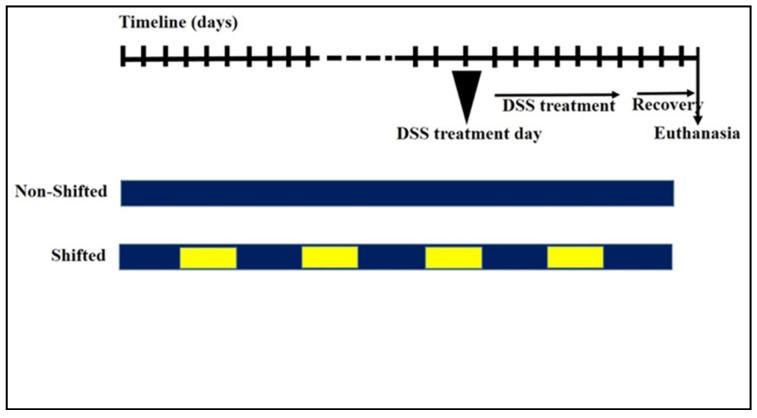
Experimental design illustrating the circadian perturbation and DSS-induced colitis protocol. Circadian perturbation was accomplished in 5-day intervals, totaling 3 months. Colitis was then induced by adding 2% DSS into the drinking water for 7 days, followed by 3 days of recovery before sacrifice. Blue: normal circadian rhythm. Yellow: shifted circadian rhythm.

**Figure 2 nutrients-16-00247-f002:**
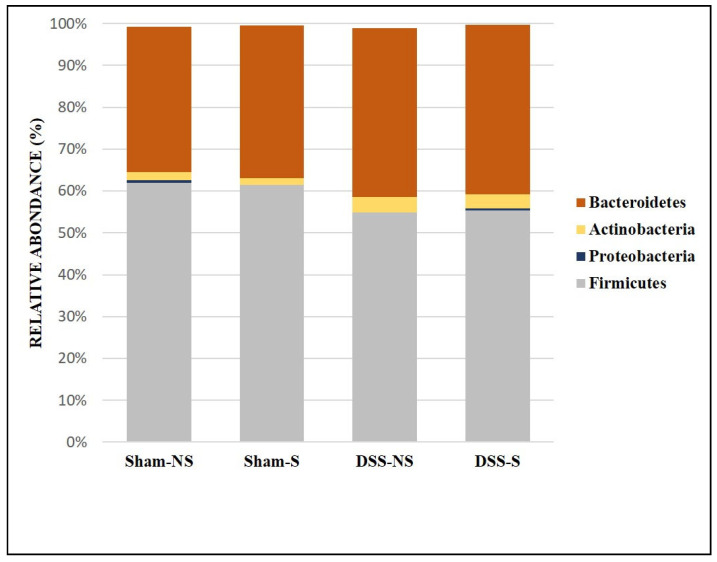
Gut microbiota colonization comparison at the phylum level of mice with normal circadian rhythm (Sham-NS), reversed circadian rhythm (Sham-S), induced colitis (DSS-NS) and induced colitis with reversed circadian rhythm (DSS-S).

**Figure 3 nutrients-16-00247-f003:**
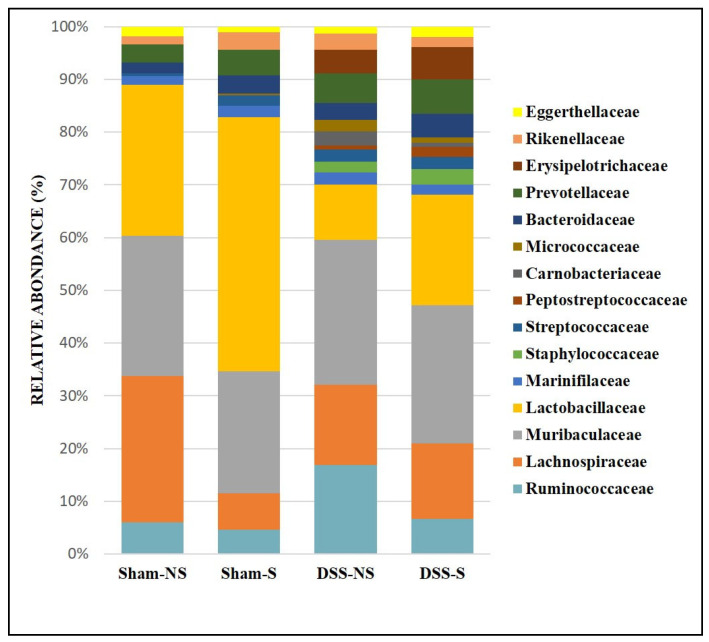
Gut microbiota colonization comparison at the family level of mice with normal circadian rhythm (Sham-NS), reversed circadian rhythm (Sham-S), induced colitis (DSS-NS) and induced colitis with reversed circadian rhythm (DSS-S).

**Figure 4 nutrients-16-00247-f004:**
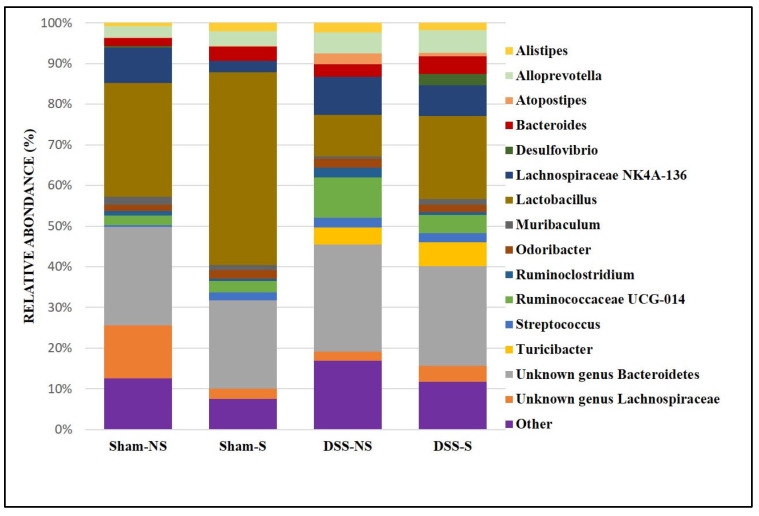
Gut microbiota colonization comparison at the genus level of mice with normal circadian rhythm (Sham-NS), reversed circadian rhythm (Sham-S), induced colitis (DSS-NS) and induced colitis with reversed circadian rhythm (DSS-S).

**Figure 5 nutrients-16-00247-f005:**
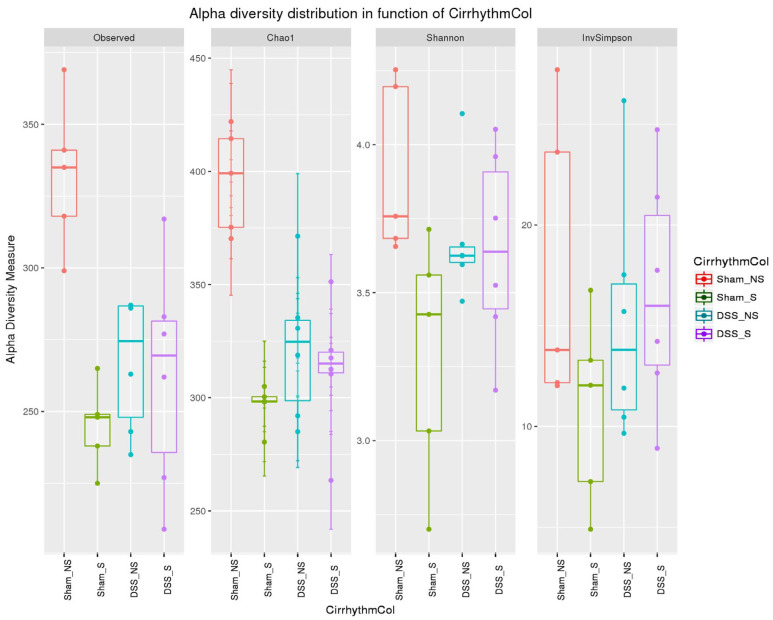
*α*-Diversity (observed OTU richness, Chao-1, Shannon diversity index and inverted Simpson index) of mice with normal circadian rhythm (Sham-NS), reversed circadian rhythm (Sham-S), induced colitis (DSS-NS) and induced colitis with reversed circadian rhythm (DSS-S).

**Figure 6 nutrients-16-00247-f006:**
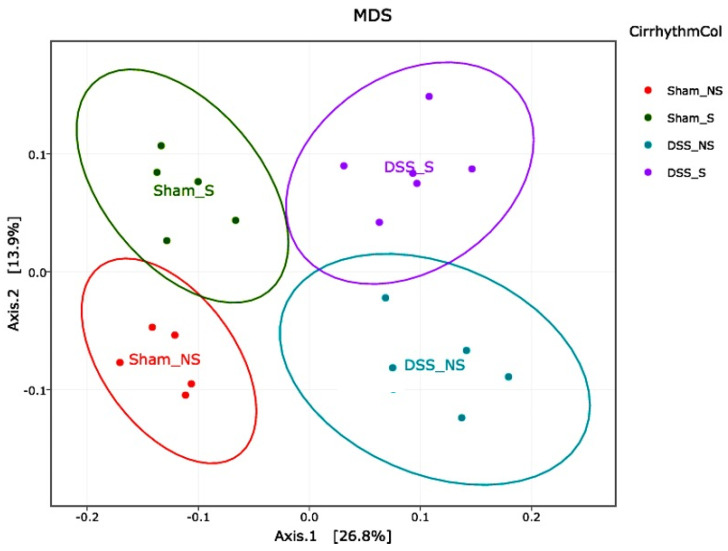
16S rRNA-based analysis demonstrates deviation from normal circadian rhythm. PCoA of mice with normal circadian rhythm (Sham-NS), reversed circadian rhythm (Sham-S), induced colitis (DSS-NS) and induced colitis with reversed circadian rhythm (DSS-S) demonstrated a separation into four distinct groups.

## Data Availability

Data are contained within the article.

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
