# Peer review of "Circadian Rhythm Perturbation Aggravates Gut Microbiota Dysbiosis in Dextran Sulfate Sodium-Induced Colitis in Mice"

_nutrients, 2024, doi:10.3390/nu16020247_

Round 1

Reviewer 1 Report

Comments and Suggestions for Authors

This is an interesting study which explores the potential effects of circadian rhythm disruption on the composition and diversity of the gut microbiota and, although its limitations, results obtained may have a broader value to better understand those mechanisms by which circadian rhythm disruption aggravates preexisting colitis. Methodology is well applied to achieve the proposed objectives, and findings here reported are consistent but poorly debated. The paper is generally well-written.

There are some minor concerns with the manuscript in its current form.

In the ABSTRACT, methodology should be rewrite, since its current form it appears that there is only one study group ("Sixty male C57BL/6J mice were exposed to circadian shifts for 3 months, and then colitis was induced by 2% DSS"). Moreover, conclusions should be moderated due to limitations identified in current study.

Although MATERIALS AND METHODS section are clearly explained and appropriate to achieve proposed objectives, detailed references should be included in "Experimental protocol", both circadian rhythm perturbation and DSS-induced colitis procedures. Moreover, quantity of fecal sample necessary for DNA extraction should be indicated.

The results are very well explained, in a continuous way, referencing the information on which the hypotheses of the work are based. The information given in the text and the figures complement each other perfectly. The figures are clear and presented in perfect order.

Regarding to DISCUSSION section, results obtained from current analysis are poorly discussed and are limited to compare results from different studies. I understand that it is very difficult comparison between the findings of the current study and those from previous studies, especially regarding fecal microbiota composition. However, it is difficult to get a clear message from the discussion. What do the results could mean in practice?

Authors should also discuss results obtained from microbial diversity analysis in the context of circadian rhythm perturbation and colitis.

Moreover, limitations should be rewritten due to information is too repetitive, and strengths for this analysis should be also included.

Others changes/questions are listed below:

1) Figure 7 should be changed!

2) "Circadian disruption alone did not induce significant changes in the physiology or GM composition. Clinical and inflammatory manifestations were significantly present only in the subgroup of mice suffering from DSS-induced colitis." Has this information been obtained from previous work by the group or other authors? Please include the required references.

3) "Bacterial expression is also influenced by light exposure: Bacteroidetes peak several hours after the beginning of the dark phase and Firmicutes peak around the beginning of the light phase [43]". Were all mice, irrespective of study group, sacrificed at the same time of day? Please, include this information in methodology section.

Author Response

We would like to thank the reviewers for their insightful comments and suggestions. We have attempted to address all their concerns in this revised paper. All corrections and additions were highlighted in yellow in the manuscript.

Please find below point by point answers to your questions.

Comment 1: In the ABSTRACT, methodology should be rewrite, since its current form it appears that there is only one study group ("Sixty male C57BL/6J mice were exposed to circadian shifts for 3 months, and then colitis was induced by 2% DSS"). Moreover, conclusions should be moderated due to limitations identified in current study.

Answer 1: The methodology in the abstract was rewritten as follow (Highlighted in yellow in the text of the abstract): “Sixty male C57BL/6J mice were divided initially to two groups, the shifted group (n=30) exposed to circadian shifts for three months and the non-shifted group (n=30) under a normal light-dark cycle. The mice of the shifted group were cyclically housed for five days under the normal 12:12 hour light-dark cycle followed by another five days under reversed light-dark cycle. At the end of the three months a colitis was induced by 2 % DSS given in the drinking water of 30 mice. Animals were then divided into four groups (n=15 per group), sham group non-shifted (Sham-NS), sham group shifted (Sham-S) and DSS non-shifted (DSS-NS) and DSS shifted (DSS-S).

In addition, the conclusion was reworded in view of the study limitations, as follow (highlighted in yellow in the text of the abstract): “These findings shows a possible interplay between circadian rhythm disruption, GM dynamics, and colitis pathogenesis”.

Comment 2: “Although MATERIALS AND METHODS section are clearly explained and appropriate to achieve proposed objectives, detailed references should be included in "Experimental protocol", both circadian rhythm perturbation and DSS-induced colitis procedures. Moreover, quantity of fecal sample necessary for DNA extraction should be indicated.

Answer 2: References were added with some added specifications; e.g. the quantity of the fecal sample (100 mg to 150 mg).

References added:

  1. Amara, J.; Saliba, Y.; Hajal, J.; Smayra, V.; Bakhos, J.J.; Sayegh, R.; Fares, N. Circadian rhythm disruption aggravates DSS-induced colitis in mice with fecal calprotectin as a marker of colitis severity. Dig Dis Sci. 2019, 64, 3122-3133.https://doi.org/10.1007/s10620-019-05675-7.
  2. Summa, KC.; Voigt, RM.; Forsyth, CB.; Shaikh, M.; Cavanaugh, K.; Tang, Y.; Vitaterna, MH.; Song, S.; Turek, FW.; Keshavarzian, A .Disruption of the Circadian Clock in Mice Increases Intestinal Permeability and Promotes Alcohol-Induced Hepatic Pathology and Inflammation. PLoS One. 2013, 8(6), e67102. https://doi.org/10.1371/journal.pone.0067102.
  3. Cooper, HS.; Murthy, SN.; Shah, RS.; Sedergran, DJ. Clinicopathologic study of dextran sulfate sodium experimental murine colitis. Lab Invest 1993, 69(2), 238–249.
  4. Perse M.; Cerar, A. Dextran sodium sulphate colitis mouse model: traps and tricks. J Biomed Biotechnol. 2012, 2012, 718617. https://doi.org/10.1155/2012/71861 7.
  5. Eichele, DD.; Kharbanda, K.K.; Dextran sodium sulfate colitis murine model: an indispensable tool for advancing our understanding of inflammatory bowel diseases pathogenesis. World J Gastroenterol. 2017, 23, 6016–6029. https ://doi.org/10.3748/wjg.v23.i33.6016. 

Comment 3: The RESULTS are very well explained, in a continuous way, referencing the information on which the hypotheses of the work are based. The information given in the text and the figures complement each other perfectly. The figures are clear and presented in perfect order.

Answer 3: Thank you for your encouraging and positive comments.

Comment 4: Regarding to DISCUSSION section, results obtained from current analysis are poorly discussed and are limited to compare results from different studies. I understand that it is very difficult comparison between the findings of the current study and those from previous studies, especially regarding fecal microbiota composition. However, it is difficult to get a clear message from the discussion. What do the results could mean in practice?

Answer 4: The obtained results show, for the first time, that the inversion of the circadian rhythm and the presence of DSS-induced colitis were associated with intestinal dysbiosis. Mice with isolated disrupted circadian rhythm have a different GM composition and a significant reduction in diversity compared to sham mice. In addition, mice with both circadian rhythm inversion and DSS-induced colitis have a different composition pattern with a significant reduction in the diversity of the bacterial gut community. Our results, despite all the limitations of this study, show a relationship between the biological clock and gut microbiota. The disruption of the clock could promote inflammation by affecting the composition of the intestinal microbiota. More and more evidence in literature appears on the effect of the biological clock on inflammatory bowel disease (Tian et al. Gastroenterology Research and Practice, 2022) and gut dysbiosis (Heddes et al. Nature communications, 2022).

Comment 5: Authors should also discuss results obtained from microbial diversity analysis in the context of circadian rhythm perturbation and colitis.

Answer 5: This part has been added to the discussion concerning the microbial diversity results (Highlighted in yellow in the text of the discussion):

“The microbial diversity analysis show that reverting circadian rhythm and inducing colitis decreased bacterial richness and diversity. These results are in accordance with a Chinese study by Hu et al, where authors found a decreased microbial species richness and diversity and a shift in microbial community with a decreased proportion of Firmicutes (72)”.

  1. Hu, L.; Li, G.; Shu, Y.; Hou, X.; Yang, L.; Jin, Y. Circadian dysregulation induces alterations of visceral sensitivity and the gut microbiota in Light/Dark phase shift mice. Front Microbiol 2022, 13, 935919. doi: 10.3389/fmicb.2022.935919.

Comment 6: Moreover, limitations should be rewritten due to information is too repetitive, and strengths for this analysis should be also included.

Answer 6: Thank you for this valuable remark. this paragraph was remodeled by removing redundant information. (strikethroughs text highlighted in yellow).

The strengthens of this work was also highlighted:

“our study contributes to the broader understanding of circadian rhythm perturbation and gut dysbiosis, offering key observations that merit further investigation”;

“the robustness of our observed dysbiosis in the murine model lays the groundwork for future studies exploring the applicability of these insights in human contexts”;

“This work guide future directions for research and clinical applications, emphasizing the possible need for probiotic interventions in circadian-disrupted individuals suffering from IBD”

Comment 7: Figure 7 should be changed!

Answer 7: it was modified to (Figure 6) in the beginning of the discussion.

Comment 8: "Circadian disruption alone did not induce significant changes in the physiology or GM composition. Clinical and inflammatory manifestations were significantly present only in the subgroup of mice suffering from DSS-induced colitis." Has this information been obtained from previous work by the group or other authors? Please include the required references.

 Answer 8: Yes, this is based on our previous work: Reference 29 (highlighted in yellow in text of the discussion and in the references section).

  1. Amara, J.; Saliba, Y.; Hajal, J.; Smayra, V.; Bakhos, J.J.; Sayegh, R.; Fares, N. Circadian rhythm disruption aggravates DSS-induced colitis in mice with fecal calprotectin as a marker of colitis severity. Dig Dis Sci. 2019, 64, 3122-3133. https://doi.org/10.1007/s10620-019-05675-7.

Comment 9: "Bacterial expression is also influenced by light exposure: Bacteroidetes peak several hours after the beginning of the dark phase and Firmicutes peak around the beginning of the light phase [43]". Were all mice, irrespective of study group, sacrificed at the same time of day? Please, include this information in methodology section.

Answer 9: Yes. All mice, irrespective of study group, were sacrificed at the same time of the day (between 9 AM and 11 AM).

Reviewer 2 Report

Comments and Suggestions for Authors

Τhis is an interesting study on the role of the circadian rhythm in affecting the gut microbiome diversity in mice subjected or not to experimental colitis.

To my opinion, the finding of microbiome alterations, and almost total differentiation [see fig 6] in the 4 study-groups contains several “scientific secrets” that need further studies to be revealed.

The differentiation of the gut microbiome in the experimental animals not subjected to colitis but only to a disturbance of the rhythm - even not statistically significant - practically sounds as important - taking into consideration the unexpected increase in lactobacilli. 

This finding should be underlined in discussion to alert researchers on how meticulous we need to be in the microbiome-related experiments, regarding adherence to the timing of stool collection.

In this context, I would like [and I make this as a proposal - I understand it cannot be included in this research] to know whether there is a microbiome diversity when samples being received at different times of the 24-hour cycle or whether there is an alteration observed when samples received a few hours after feeding or after different foods e.g.  fat.

Another issue - and this should be discussed - is what would be the findings when colitis induction precedes circadian rhythm changes. In other words, does the “double hit” theory apply bidirectionally?

Also of interest would be the histological evaluation of the gut in colitis animals when established in a previous disturbed gut microbiome. In other words - given the increase in Lactobacilli in circadian rhythm affected mice – are the colitis lesions comparable? Or Lactobacilli positively affect, or protect, the gut mucosa

I dare to say that you could further enhance your research only on the role of circadian rhythm in the gut microbiome alterations, taking into consideration and other parameters possibly affected; what about hormones related to light/dark cycle disruption?  How the brain-gut axis is implicated in this process? it seems bidirectional, but is it correct?  [And to leave the colitis induction for another paper!]

MINOR REMARKS

In the 1st paragraph of Discussion, you refer to Figure 7 - I can't find it in the text. Is it Figure 6?

The 3rd paragraph of the results [2nd on page 5] "with respect to GM and ..." is rather a commentary phrase than a description of your findings, so its place, along with its literature, should be transferred to discussion.

Author Response

We would like to thank the reviewers for their insightful comments and suggestions. We have attempted to address all their concerns in this revised paper. All corrections and additions were highlighted in yellow in the manuscript.

Please find below point by point answers to your questions.

Τhis is an interesting study on the role of the circadian rhythm in affecting the gut microbiome diversity in mice subjected or not to experimental colitis.

Comment 1: To my opinion, the finding of microbiome alterations, and almost total differentiation [see fig 6] in the 4 study-groups contains several “scientific secrets” that need further studies to be revealed.

The differentiation of the gut microbiome in the experimental animals not subjected to colitis but only to a disturbance of the rhythm - even not statistically significant - practically sounds as important - taking into consideration the unexpected increase in lactobacilli. 

This finding should be underlined in discussion to alert researchers on how meticulous we need to be in the microbiome-related experiments, regarding adherence to the timing of stool collection.

Answer 1: This part has been added to the discussion concerning the finding of microbiome alterations and figure 6  (Highlighted in yellow in the text of the discussion):

“The microbial diversity analysis show that reverting circadian rhythm and inducing colitis decreased bacterial richness and diversity. These results are in accordance with a Chinese study by Hu et al, where authors found a decreased microbial species richness and diversity and a shift in microbial community with a decreased proportion of Firmicutes (72)”.

  1. Hu, L.; Li, G.; Shu, Y.; Hou, X.; Yang, L.; Jin, Y. Circadian dysregulation induces alterations of visceral sensitivity and the gut microbiota in Light/Dark phase shift mice. Front Microbiol 2022, 13, 935919. doi: 10.3389/fmicb.2022.935919.

Comment 2: In this context, I would like [and I make this as a proposal - I understand it cannot be included in this research] to know whether there is a microbiome diversity when samples being received at different times of the 24-hour cycle or whether there is an alteration observed when samples received a few hours after feeding or after different foods e.g.  fat.

Answer 2: Although the intestinal microbiota is constantly exposed to environmental challenges and different foods, its composition and function in an individual are stable against perturbations, as microbial communities are resilient and resistant to change. Receiving samples from different times of the 24-hour cycle, most probably would not affect microbiome diversity. Microbial communities are even resilient to antibiotic challenges, as intestinal microbiome reestablishes itself after antibiotic therapy.

Comment 3: Another issue - and this should be discussed - is what would be the findings when colitis induction precedes circadian rhythm changes. In other words, does the “double hit” theory apply bidirectionally?

Answer 3: It will be very interesting to demonstrate this bidirectional interplay; however, it will be difficult experimentally to create such a model:  the validated circadian rhythm inversion protocol is of 3 months, and it will be hard to induce colitis by DSS and maintain a chronic inflammatory state in mice for more than 3 months.

Comment 4: Also, of interest would be the histological evaluation of the gut in colitis animals when established in a previous disturbed gut microbiome. In other words - given the increase in Lactobacilli in circadian rhythm affected mice – are the colitis lesions comparable? Or Lactobacilli positively affect, or protect, the gut mucosa

Answer 4: In our previous work (reference 29) we have showed that the histological index was more elevated in the DSS-group compared to sham, and especially in the DSS-S group where significant histological findings, reflecting the severity of histological involvement like: cryptic abscess, length of the colon and mucosal thickening were found.

Comment 5: I dare to say that you could further enhance your research only on the role of circadian rhythm in the gut microbiome alterations, taking into consideration and other parameters possibly affected; what about hormones related to light/dark cycle disruption?  How the brain-gut axis is implicated in this process? it seems bidirectional, but is it correct?  [And to leave the colitis induction for another paper!].

Answer 5: You are right, but this was not our primary objective in this study. The relation gut-brain axis is well established in irritable bowel syndrome, however further studies are needed to elucidate better this role in IBD.

MINOR REMARKS

Comment 6: In the 1st paragraph of Discussion, you refer to Figure 7 - I can't find it in the text. Is it Figure 6?

Answer 6: You are right. It was modified accordingly.

Comment 7: The 3rd paragraph of the results [2nd on page 5] "with respect to GM and ..." is rather a commentary phrase than a description of your findings, so its place, along with its literature, should be transferred to discussion.

Answer 7: Indeed, this paragraph was transferred to the discussion section (highlighted in yellow).

We hope the reviewers will find our paper now suitable for publication.

Kind regards

Corresponding authors